# Multiple Na,K-ATPase Subunits Colocalize in the Brush Border of Mouse Choroid Plexus Epithelial Cells

**DOI:** 10.3390/ijms22041569

**Published:** 2021-02-04

**Authors:** Inga Baasch Christensen, Lei Cheng, Jonathan R. Brewer, Udo Bartsch, Robert A. Fenton, Helle H. Damkier, Jeppe Praetorius

**Affiliations:** 1Department of Biomedicine, Faculty of Health Science, Aarhus University, 8000 Aarhus, Denmark; ibch@biomed.au.dk (I.B.C.); lche@biomed.au.dk (L.C.); robert.a.fenton@biomed.au.dk (R.A.F.); hd@biomed.au.dk (H.H.D.); 2Department of Biochemistry and Molecular Biology, Faculty of Science, University of Southern Denmark, 5230 Odense, Denmark; brewer@memphys.sdu.dk; 3Department of Ophthalmology, Experimental Ophthalmology, University Medical Center Hamburg-Eppendorf, 20246 Hamburg, Germany; u.bartsch@uke.uni-hamburg.de

**Keywords:** Na,K-ATPase, choroid plexus, cytoskeleton, membrane targeting, membrane stabilization, actin, ankyrin, spectrin

## Abstract

(1) Background: The unusual accumulation of Na,K-ATPase complexes in the brush border membrane of choroid plexus epithelial cells have intrigued researchers for decades. However, the full range of the expressed Na,K-ATPase subunits and their relation to the microvillus cytoskeleton remains unknown. (2) Methods: RT-PCR analysis, co-immunoprecipitation, native PAGE, mass spectrometry, and differential centrifugation were combined with high-resolution immunofluorescence histochemistry, proximity ligase assays, and stimulated emission depletion (STED) microscopy on mouse choroid plexus cells or tissues in order to resolve these issues. (3) Results: The choroid plexus epithelium expresses Na,K-ATPase subunits α1, α2, β1, β2, β3, and phospholemman. The α1, α2, β1, and β2, subunits are all localized to the brush border membrane, where they appear to form a complex. The ATPase complexes may stabilize in the brush border membrane via anchoring to microvillar actin indirectly through ankyrin-3 or directly via other co-precipitated proteins. Aquaporin 1 (AQP1) may form part of the proposed multi-protein complexes in contrast to another membrane protein, the Na-K-2Cl cotransporter 1 (NKCC1). NKCC1 expression seems necessary for full brush border membrane accumulation of the Na,K-ATPase in the choroid plexus. (4) Conclusion: A multitude of Na,K-ATPase subunits form molecular complexes in the choroid plexus brush border, which may bind to the cytoskeleton by various alternative actin binding proteins.

## 1. Introduction

The choroid plexus (CP) includes a highly active transporting epithelium located in the ventricular system of the brain, where it secretes the majority of the cerebrospinal fluid (for review [1]). The CP epithelium (CPE) consists of a cellular monolayer, facilitating the selective transport of ions, solutes, and water into the lumen of the brain ventricles. Microvilli and basolateral infoldings in the plasma membrane of the CPE cells (CPECs) vastly increase the area available for the concerted actions of various transporter proteins and channels [2].

The CPECs display an atypical luminal-basolateral distribution of certain transport proteins, as compared to almost all other epithelia. Most strikingly, the Na,K-ATPase is expressed in the brush border membrane of CPECs while it is widely recognized as a basolateral plasma membrane protein in most other epithelial cells. The Na,K-ATPase complex consists of an α subunit containing the catalytic site for ATP hydrolysis and ion translocation, and a β subunit responsible for the plasma membrane targeting of the protein complex. The Na,K-ATPase α1 and β1 subunits are ubiquitously expressed, whereas other subunit isoforms are cell type specific (for review [3]). Analyses of Na,K-ATPase subunits at the mRNA and protein levels consistently suggest the expression of α1, β1, and β2 in the CP, whereas expression of α2–4 and β3 is unclear [4,5,6]. The CPECs also express the Na,K-ATPase γ-subunit FXYD1 [7]. In CPECs, the Na,K-ATPase α1, β1, and β2 subunits have all been localized to the luminal membrane [4,8,9]. Relevant to the current context, the molecular expression of the Na,K-ATPase β2 subunit has been studied in connection to luminal membrane accumulation of Na,K-ATPases in cell cultures [10,11].

When expressed in the basolateral membrane, the Na,K-ATPase is anchored to the actin cytoskeleton through the anchoring protein ankyrin and the spectrin network [12,13]. Although the accumulation of the Na,K-ATPase in the luminal membrane of CPECs was studied previously in the relation to both ankyrin and spectrins there was no clear indication on why these proteins colocalize with the Na,K-ATPase in the brush border [14,15,16]. These studies did, however, reveal an independence of the opposing accumulation of the luminal Na,K-ATPase and the basolateral AE2 on the microtubular network [14], and that the expression of B-cadherin in chicken CP may have implications for the luminal accumulation of the Na,K-ATPase [16]. The expression of B-cadherin seems to be insufficient at causing the luminal accumulation of Na,K-ATPases, as the same cadherin does not prevent the basolateral expression of Na,K-ATPase in absorptive epithelia, as e.g., the intestine. Basically, luminal membrane accumulation of Na,K-ATPase in the CP requires a delivery system/machinery to bring the proteins to the opposite membrane than usual, as well as a mechanism to stabilize the proteins in the luminal membrane. Luminal membrane accumulation of the Na,K-ATPase might occur as a result of direct trafficking from the trans-Golgi network, from selective internalisation and degradation of basolateral complexes, or from transcytosis from the basolateral to the brush border membrane. The stabilization of membrane proteins delivered to the luminal membrane must be accomplished by anchoring them selectively to the brush border cytoskeleton or by immobilization through protein interactions within the plasma membrane. In the CPECs, ankyrin-3 is localized predominantly to the luminal membrane domain, where some spectrin immunoreactivity was also found [14,15,16]. The localization of ankyrin and spectrins was consistent with a role in stabilizing the Na,K-ATPase complexes in the luminal membrane of CPECs. However, we have previously compared the subcellular localization of ankyrin-3 and spectrins in the CPE and found that the spectrin immunoreactivity did not appear in the microvilli [15]. 

The current study was undertaken in order to further explore the mechanisms in the luminal accumulation/stabilization of Na,K-ATPase complexes in CPECs. We wished to determine whether expression of Na,K-ATPase β2 is necessary for luminal accumulation of the Na,K-ATPase complex, whether the Na,K-ATPases in the luminal membrane associates with alternative anchoring or cytoskeletal proteins, or whether other membrane proteins are associated with the Na,K-ATPase in the membrane or en route to their final position in CPECs. We employed several methods to identify candidate protein interaction partners of the Na,K-ATPase α1 subunit, determined whether they form distinct sub-cellular structures, and determined where these putative interactions might take place in the cells. Our major findings are: (1) That Na,K-ATPase β2 expression is not necessary for luminal accumulation of the Na,K-ATPase complex, (2) that the multiple Na,K-ATPase subunits seem to migrate as one complex with AQP1 but not NKCC1, the Na-dependent chloride-bicarbonate exchanger (Ncbe) or the anion exchanger 2 (AE2), (3) that both NKCC1 and the Na,K-ATPase α1 subunit co-colocalize with AE2 and Ncbe in intracellular structures, (4) that the Na,K-ATPase complex may anchor to the actin cytoskeleton via either ankyrin-3 or alternative intermediate proteins identified by co-immunoprecipitation and mass spectrometry, and finally (5) that complete luminal membrane accumulation of Na,K-ATPase subunits and AQP1 in CPECs require NKCC1 expression suggesting that basolateral delivery of these predominantly luminal proteins occur in the CP.

## 2. Results

### 2.1. The CP Expresses Several Na,K-ATPase Subunits in the Brush Border

Figure 1A exemplifies RT-PCR analysis of the expression of mRNAs encoding Na,K-ATPase subunits in FACS isolated CPECs. The data demonstrate the expression of α1, α2, β1, β2, and β3 Na,K-ATPase subunits in these cells, while the expression of α3 and α4 Na,K-ATPase subunits was not observed by this method. Co-immunoprecipitation using anti-Na,K-ATPase α1 antibodies and mass spectrometry identified α1, α2, α4, β1, β2, and β3 subunits along with the γ-subunit FXYD1 (Table 1). Mass spectrometry of bands cut out of native gels detected the same subunits as well as the Na,K-ATPase α3 subunit (Table 2, Appendix A). The subcellular localization of Na,K-ATPase α2 and β2 to the luminal membrane domains is shown in Figure 1B–D. Staining was performed with two different Na,K-ATPase β2 antibodies because of the weak signal. STED microscopy verified the brush border expression of Na,K-ATPase α1, α2, and β1 subunits (Appendix A). Quantitation of the co-localization of Na,K-ATPase α1 with α2 subunits and Na,K-ATPase α1 with β1 subunits are similarly to the positive control Na,K-ATPase β1 with β1 subunits (Appendix A, Table 3). Analysis of over-exposure micrographs after masking the robust brush border signal, indicate that a minor fraction of Na,K-ATPase α1 can be detected in the perinuclear region of the epithelial cells, whereas no basolateral staining was observed in CPECs from normal mice (Figure 1E,F).

The proximity of Na,K-ATPase α1 to Na,K-ATPase β1 and Na,K-ATPase α2 subunits was verified by PLA (Figure 2). Figure 2A–C illustrates that Na,K-ATPase α1 reaction signal with Na,K-ATPase β1 was observed mainly at the luminal membrane domain including the brush border. A similar reaction pattern was observed using anti-Na,K-ATPase α1 with anti-Na,K-ATPase α2 antibodies (Figure 2D–F). Positive control PLA analysis using only Na,K-ATPase α1 antibody and both “+” and “−” probes targeting mouse IgG showed brush border reactions (Appendix A). By contrast, negative control PLA using the same Na,K-ATPase α1 antibody and only “+” probe was devoid of signal (Appendix A). Thus, several lines of evidence suggest the close proximity of more than the canonical of Na,K-ATPase α1 and Na,K-ATPase β1 subunits in the luminal membrane. However, it is uncertain whether or not Na,K-ATPase α3, α4, or β4 are expressed in the CPECs as it was not possible to immunolocalize these subunits.

### 2.2. Na,K-ATPase α1 Immunoprecipitates Both Luminal and Basolateral Transport Proteins 

Little is known regarding the protein–protein interactions that may be involved in the unusual accumulation and stabilization of Na,K-ATPase complexes in the brush border membrane of the CP. We applied an anti-Na,K-ATPase α1 antibody for co-immunoprecipitation experiments followed by mass spectrometry of the eluates of CP proteins in order to identify candidate proteins that interacted with the Na,K-ATPase complex. Table 1 summarizes the array of plasma membrane, anchoring, and cytoskeletal proteins detected. As mentioned above, the anti-Na,K-ATPase α1 antibody co-precipitated several other Na,K-ATPase subunits including FXYD1. A control immunoblot verified the co-immunoprecipitation of Na,K-ATPase β1 and probably ankyrin-3 with Na,K-ATPase α1 (Appendix A). Conversely, ankyrin-3 antibodies only precipitated ankyrin-3 and proteasome 20s antibodies did not precipitate any of the other proteins. Interestingly, the luminal membranes AQP1, NKCC1, and the inward rectifier potassium channel 13 (Kir7.1) were also detected in the analysis as well as the basolateral Ncbe and AE2. These data suggest that a fraction of the cellular pool of Na,K-ATPase α1 converge with other luminal and even basolateral transport proteins at some point in the biosynthetic pathway or en route to the luminal membrane. This possibility was further examined using differential centrifugation of CP protein samples. Figure 3A,B shows the results of immunoblotting various subcellular fractions against a selected array of proteins. All the transport proteins included in the analysis were most robustly detected in the densest fractions, which are most likely to represent the plasma membrane fraction and/or sample components failing to enter the gel (f16). In contrast, the example soluble protein ankyrin-3 was most abundant in lighter fractions (f5–f8) as well as in fraction f16 (Figure 3B). Exclusion of the f16 fraction from the analysis revealed differences in the distribution of the intracellular pools of membrane proteins (Figure 3C). While the Na,K-ATPase subunits α1, α2, and β1 have more similar distribution to AQP1 than to AE2 and NKCC1, it is not possible from these analyses to suggest a subcellular compartment for the proposed interactions. This can be better achieved by mass spectrometry analysis of co-migrating native proteins in gels and by proximity ligation assays.

### 2.3. Na,K-ATPases and AQP1 Exist in the Same Subcellular Fractions

The native gel mass spectrometry gel cut-out analysis (Appendix A, Table 2, Table 4 and Table 5) and immunoblot analysis (Appendix A) suggested that Na,K-ATPase α1 co-migrates with other Na,K-ATPase subunits, as well as AQP1, and ankyrin-3. As described above, AQP1 was convincingly co-immunoprecipitated by the Na,K-ATPase α1 antibody and was detected with a similar distribution by gradient centrifugation. Most of the physical proximity may arise from the extensive expression of the proteins in the brush border. We verified the predominant brush border colocalization of Na,K-ATPases and AQP1 by STED microscopy (Appendix A, Table 3). Furthermore, the very sensitive PLA produced the expected luminal signal when anti-Na,K-ATPase α1 and anti-AQP1 antibodies were applied (Figure 4A–C). A signal was, however, also detected intracellularly, but to a much lesser extent.

The proteins identified in the Na,K-ATPase enriched band from native gels also suggested an association of the complex to both the plasma membrane associated anchoring proteins and the actin cytoskeleton, as well as to the intracellular trafficking machinery by association to the microtubule network (Table 2). Both actin and the tubulins were also among the proteins that were co-immunoprecipitated repeatedly by the anti-Na,K-ATPase α1 antibody (Table 1). Figure 5A,B illustrates the robust expression of β-actin along the entire length of the microvilli of the CP brush border. A robust PLA signal was observed in the brush border area also when analyzing actin and Na,K-ATPase α1 colocalization (Figure 5C–E). Thus, it is likely that Na,K-ATPase subunits and AQP1 co-exist in close proximity in the luminal membrane close to β-actin and intracellular structures associated to microtubules.

### 2.4. NKCC1, Ncbe, and AE2 Belong to the Same Subcellular Fractions

As described above, the Na,K-ATPase α1 antibody co-immunoprecipitated NKCC1, as well as the basolateral AE2 and Ncbe (Table 1). PLA analysis was performed in order to visualize the most probable cellular sites for the interaction of NKCC1 and Na,K-ATPase α1 subunits. Figure 6 exemplifies analysis of the proximity of Na,K-ATPase α subunits and NKCC1. The Na,K-ATPase α1 antibody produced reaction products with the NKCC1 antibody almost exclusively in the brush border of the CPECs (Figure 6A–C). A similar reaction pattern was observed with Na,K-ATPase α2 and NKCC1 antibodies, although with a much weaker signal (Figure 6D–F). Positive control PLA using only NKCC1 antibody and both “+” and “−” probes targeting rabbit IgG showed predominantly brush border signal (Appendix A). Although the Na,K-ATPase and NKCC1 are predominantly expressed in the brush border of the CPE, the native gel immunoblot analysis and mass spectrometry gel cut-out analysis (Appendix A, Table 2, Table 4 and Table 5) indicated that the Na,K-ATPase complexes and NKCC1 containing structures do not co-migrate in gels. Unlike AQP1 and Kir 7.1, the site of putative interaction of Na,K-ATPase complexes and NKCC1 seem not to be in the ankyrin-3 positive band of the native gels (Appendix A, Table 2, Table 4 and Table 5). A pattern with predominant brush border PLA reaction was also observed using anti-AQP1 and -NKCC1 antibodies (Figure 6G–I), although these proteins did not co-migrate in native gels. Therefore, the PLA reactions in the brush border of the CPECs merely indicates proximity but may not suggest actual physical interactions.

NKCC1 co-migrated in native gels with the luminal protein TRPv4 as well as with basolateral proteins such as Ncbe and AE2 (Table 4 and Table 5). The seemingly paradoxical finding may indicate the co-existence of the luminal and basolateral proteins within the biosynthetic pathway or in the trafficking machinery. Indeed, PLA analysis of NKCC1, AE2, and Ncbe produced mainly perinuclear signals (Figure 7A–F). Appendix A shows positive controls using only the anti-Ncbe antibody with an anti-rabbit “+” and “−” secondary antibody probes. The individual spots of reaction products were not resolved for these control PLAs. Taken together, these data suggest that the co-migration of NKCC1 with AE2 and Ncbe in native gel cut-outs reflects co-existence in intracellular compartments rather than in the plasma membrane. In contrast to the Na,K-ATPase and AQP1, the NKCC1 and the basolateral membrane proteins Ncbe and AE2 seem to co-migrate with spectrin forms and β-actin.

### 2.5. Intracellular Proximity of a Subset of Na,K-ATPase Subunits with AE2 and Ncbe

A perinuclear pattern of PLA signal was observed when anti-NaK-ATPase α1 was combined with either anti-AE2 or anti-Ncbe antibodies (Figure 7G–L). These reaction products indicate that NaK-ATPase α1 only obtain proximity with the two bicarbonate transporters in intracellular structures. This might explain the results from co-immunoprecipitation experiments, as PLA is extremely sensitive. However, the native gel results indicate that these are rarer events than the macromolecular structures that co-migrate in the native gel, where the bicarbonate transporters and NKCC1 migrate at higher molecular weight complexes compared to the Na,K-ATPases. The electrogenic Na-bicarbonate cotransporter 2 (NBCe2) is a luminal membrane protein in the CP, that was not co-immunoprecipitated by the anti-Na,K-ATPase α1 antibody and was not detected in the native gel cut-outs. Figure 8A–C shows that anti-NBCe2 only produced a minor reaction signal with the anti-NaK-ATPase α1 antibody. This sparse signal was mainly observed in the perinuclear zone of the cells. It is noteworthy that there is very little—if any—reaction products corresponding to the brush border, as both proteins are robustly expressed in this domain but unlikely to interact physically. This also suggests a high degree of reliability of the method apparently producing very limited false positive reaction signals. By contrast, PLA reaction products were occasionally observed near the base of the microvilli for anti-NBCe2 and anti-NKCC1 antibodies (Figure 8D–F). Positive and negative controls for proximity analysis are shown in Appendix A for anti-NBCe2, where the PLA signal required the presence of both “+” and “−” secondary antibody probes. For comparison, Appendix A summarizes the distribution of PLA products for all applied combinations of primary antibodies and controls. Thus, separation of the Na,K-ATPase and NKCC1 in analyses of the density gradient fractions and in native gel cut-outs may indicate that Na,K-ATPase α1 antibodies co-immunoprecipitates NKCC1 only because of indirect protein interactions in the brush border. Both the Na,K-ATPase α1 subunit and NKCC1 can be detected at intracellular sites in close proximity to the basolateral proteins AE2 and Ncbe by PLA, but for the Na,K-ATPase these structures are not exposed by the analysis of native gel cut-outs.

### 2.6. Brush Border Proximity of Na,K-ATPase α1 Subunits and Ankyrin-3 in the CP

Conventionally, Na,K-ATPase complexes are anchored to the actin cytoskeleton via their association to ankyrin and spectrin as mentioned in the introduction. In the CPE, ankyrin-3 immunoreactivity was clearly detectable in the brush border area of CPECs, probably mainly in the basal half of the microvilli (Figure 9A,B). Ankyrin-3 was also detected by immunoblotting of Na,K-ATPase α1 co-immunoprecipitates and the proteins co-migrated on native gels as mentioned above (Appendix A, respectively). Density gradient separation of subcellular structures further revealed enrichment of Na,K-ATPase subunits and ankyrin-3 in both light fractions (f5–8) and the heavy plasma membrane fraction (f16, Figure 3). This indicates that the Na,K-ATPase complex interacts with ankyrin-3 at both intracellular sites and at the plasma membrane as supported by PLA analysis of ankyrin-3 and Na,K-ATPase α1, α2 or β1 subunits (Figure 10). Indeed, ankyrin-3 produced a robust signal with Na,K-ATPase α1 corresponding to the brush border of the cells and a sparser intracellular signal (Figure 10A–C). Interestingly, the anti-ankyrin-3 reaction products were predominantly intracellular with both of anti-Na,K-ATPase α2 and anti-Na,K-ATPase β1 antibodies (Figure 10D–I). This may result from a pool of Na,K-ATPase complexes residing in the plasma membrane anchoring to ankyrin-3 through the Na,K-ATPase α1 subunit and that the other subunits obtain closer proximity to the ankyrin-3 only in intracellular structures. Control PLA using ankyrin-3 and NKCC1 antibodies produced a reaction pattern that was indistinguishable from the one produced by ankyrin-3 and the Na,K-ATPase α1 subunit (Figure 10J–L). The majority of the signal was observed in the brush border area, while few products were consistently observed inside the cells, suggesting that brush border NKCC1 is anchored in a similar fashion as Na,K-ATPase α1 presumably via ankyrin-3.

### 2.7. Anchoring of the Brush Border Na,K-ATPase α1 Subunits May Not Involve the Spectrin Network

Several spectrin forms have been detected previously in the CP, but the association of Na,K-ATPase complexes with conventional spectrin forms has not been shown. The observation that Na,K-ATPase α1 co-immunoprecipitated α2-spectrin in one of five samples (Table 1) seems, to some degree, to contradict our previous observation of a spatial distance in the luminal membrane domain of spectrin and the brush border Na,K-ATPase. However, several lines of evidence indicate minimal interaction between spectrin forms and the Na,K-ATPase complex. Firstly, we verified the expression of α2-, β1-, β2-, and β3-spectrins in the CP by RT-PCR analysis on FACS purified epithelial cells (Appendix A). The first three of these can be readily immunolocalized in the CP, and only α2- and β2-spectrin are associated with the luminal membrane domain. In contrast to ankyrin-3 and actin, neither α2- nor β2-spectrin staining was detectable in the length of brush border of the CPECs (Figure 11A,B). The labeling was abundant only in the basolateral membrane including the basal labyrinth and just beneath the luminal brush border membrane corresponding to the terminal web. Double-fluorescence immunohistochemistry with Na,K-ATPase α1 supported the notion that the luminal domain spectrins are localized just beneath the brush border and rarely colocalize with the Na,K-ATPase (Figure 11C,D). 

Figure 11E–G show PLA analysis of α2-spectrin with Na,K-ATPase α1. The reaction products were very infrequently detected and apparently not corresponding to the microvilli. Similar analysis with α2-spectrin and ankyrin-3 also revealed sparse reaction products, and only in the luminal membrane domain beneath the brush border (Figure 11H–J). Further evidence of infrequent anchoring of Na,K-ATPases via spectrin in the CP brush border was the relatively minor colocalization of one spectrin and Na,K-ATPase subunits as compared to actin and Na,K-ATPase subunits by STED microscopy (Appendix A). Finally, spectrin forms were not detected in the mass spectrometry analysis of native gel cut-out samples containing Na,K-ATPase subunits but were readily identified in the higher molecular weight bands by mass spectrometry (Table 4 and Table 5). These higher molecular weight complexes contained NKCC1, AE2, and Ncbe and thus most likely represents plasma membrane fractions or structures associating with spectrin en route to the membrane. Thus, the Na,K-ATPase in the CP brush border might link to the actin cytoskeleton of the microvillus core via ankyrin-3 and another actin binding protein than spectrin.

Alternative actin-binding proteins detected by Na,K-ATPase α1 co-immunoprecipitation and mass spectrometry include annexin-A2, junction plakoglobin (γ-catenin), β1- and β2-syntrophin, gelsolin, and α-actinin (Table 1). Annexin-2, which belongs to the spectrin gene family, was identified more frequently than conventional spectrins by co-immunoprecipitation (Table 1). Thus, annexin-2 might be the missing link in the anchoring of Na,K-ATPases to the microvillar core in the CPE brush border. It is not possible to rule out other proteins for this function, as we were unable to obtain suitable antibodies for immunolocalization of the candidate actin binding proteins.

### 2.8. Partial Basolateral Accumulation of NaK-ATPase and AQP1 in NKCC1 ko CP

Correct membrane targeting and specific accumulation of the Na,K-ATPase complexes may be dependent on the interim co-existence of transport proteins destined for both the luminal and basolateral plasma membranes in intracellular structures. Appendix A illustrates that there are no gross changes in the expression levels of transporter proteins in mouse models of genetic disruption of NKCC1, AQP1, the Na-hydrogen-exchanger 1 (NHE1), or the electroneutral Na-bicarbonate cotransporter 1 (NBCn1) as assessed by semi-quantitative immunofluorescence histochemistry. This is similar to our findings in a β2/β1 Na,K-ATPase knock-in mouse model. The only significant changes in transporter protein expression were an increased expression of Na,K-ATPase α1 and NKCC1 and a decrease of Na,K-ATPase α2 in NBCn1 ko mouse CP compared to wildtype littermates (Appendix A, *n* = 5). Interestingly, homozygous Na,K-ATPase β2/β1 knock-in CP displayed a similar Na,K-ATPase β1 expression level as in the β2/β1 heterozygous tissue (Appendix A), indicating that lack of Na,K-ATPase β2 expression does not change the expression levels or the membrane targeting of Na,K-ATPase β1. This suggests that luminal accumulation of Na,K-ATPase complexes in the CPECs does not rely on the heavily glycosylated Na,K-ATPase β2.

Immunohistochemical analysis revealed a striking redistribution of Na,K-ATPase α1, α2, and β1 subunits as well as AQP1 in the NKCC1 knockout model (Figure 12). Figure 12A illustrates the CP cellular immunoreactivity for Na,K-ATPase α1 in NKCC1 wildtype mice, whereas Figure 12B illustrates its typical distribution in NKCC1 knockout mice. The partial basolateral accumulation of Na,K-ATPase α1 subunit immunoreactivity was most pronounced corresponding to the basolateral labyrinth (infoldings). Histograms of line-scans of the cross-cellular staining intensities of Na,K-ATPase subunits from similar micrographs are shown in Figure 12C–E and for AQP1 in Figure 12F. Significantly elevated labeling intensities were observed in the basal part of CPECs (centered at bin#10) for all four proteins. Basolateral redistribution of Na,K-ATPase subunits and AQP1 was not observed in any of the other genetically modified mouse models. This is exemplified by the lack of basolateral accumulation of Na,K-ATPase β1 subunits in the AQP1 knockout mouse (Appendix A). Neither NBCe2 nor Ncbe redistributed in the NKCC1 knockout model (Appendix A). 

## 3. Discussion

The expression of the Na,K-ATPase in the luminal membrane of the CP has consistently puzzled researchers ever since its discovery [17,18]. 

Several approaches have been applied unsuccessfully to determine whether the luminal expression was dependent on (1) direct targeting of the Na,K-ATPase complex to the luminal membrane, or (2) selective stabilization of the complex in the brush border membrane. The conventional Na,K-ATPase α1 and β1 are expressed along with the γ-subunit FXYD1 in this tissue [7,17,19,20,21], which would usually result in basolateral expression of the complex. The protein complexes have been shown to anchor to the cytoskeleton through binding of Na,K-ATPase α1 to ankyrin and through spectrin to actin in other cell types [13,16]. 

It is very clear from the literature that CSF secretion as such is driven by Na^+^ extrusion across the luminal membrane and that this is accomplished directly by Na,K-ATPase activity in the CP as reviewed previously [1]. However, it is unclear why the CP is not equipped with a simple configuration of Na,K-ATPase α1/β1/FXYD1 subunits. One explanation for the expression of multiple Na,K-ATPase subunits might be the embryological origin of the CPE as for other neuroepithelia. The expression of the various subunits may also change during development and maturation and serve different purposes at different stages. In addition, some redundance of this critical enzyme complex might also be beneficial for surviving challenges to the CSF secreting system.

It has been speculated that luminal membrane accumulation of the Na,K-ATPase results from expressing alternative subunits to the α1 and β1 subunit. Specifically, the expression of Na,K-ATPase β2 and β3 subunits, which possess larger and negatively charged glycans, has been reported to stabilize Na,K-ATPase complexes in the luminal membrane [11]. Indeed, several laboratories have reported expression of more subunits in this tissue. Na,K-ATPase β2 as well as β3 subunit expression have been reported in either the intact tissue or in FACS isolated CPECs [4,5,6]. Here, we present evidence that the CPECs express at least α1, α2 with β1, β2, and β3 subunits at the mRNA level. Furthermore, Na,K-ATPase α2 and β2 were immunolocalized to the brush border membrane along with α1, β1, and FXYD1. 

Our data suggest that a small fraction of Na,K-ATPase α1, α2, and β2 seem to reside inside cytosolic compartments as assessed by over-exposure IHC, STED microscopy, and PLA. This is consistent with previous assessment of the Triton-X soluble and insoluble fractions of the Na,K-ATPase in the CP indicating more than 70% of the complex to be tightly associated with the cytoskeleton [16]. AQP1 and NKCC1 are like the Na,K-ATPase complexes robustly expressed in the brush border membrane, while intracellular pools of the proteins were not readily detectable in the CP by conventional IHC. Because of the predominant brush border expression of these proteins the vast majority of Na,K-ATPase and AQP1 or NKCC1 PLA signal was observed corresponding to the brush border, although perinuclear reaction products were observed less frequently. Conversely, AE2 and Ncbe were immunolocalized selectively at the basolateral plasma membrane. Nevertheless, PLA allowed the detection of intracellular sites of close proximity between the luminal and basolateral proteins, exemplified by reaction signals between Na,K-ATPase subunits and AE2 and Ncbe. Intracellular sites of protein proximity were also found for AQP1 or NKCC1 with AE2 and Ncbe. The existence of such intracellular fractions was supported by mass spectrometry identification of all of these proteins after Na,K-ATPase α1 co-immunoprecipitaiton. Furthermore, cut-outs of clearly defined bands from Coomassie stained native PAGE gels contained either Na,K-ATPase subunits with AQP1 or NKCC1 with AE2 and Ncbe. Analysis of density gradient fractions of CP homogenates yielded a similar pattern and is consistent with previous identification of separate association patterns of e.g., Na,K-ATPase and AE2 [14,16]. These fractions would either reflect specific structures within the biosynthetic pathway for the membrane transporters or co-migrating mixed luminal and basolateral membrane, as the NKCC1 is a brush border protein while AE2 and Ncbe are basolateral proteins. 

The lack of Na,K-ATPase β2 expression in the CP of homozygous Na,K-ATPase β2/β1 knock-in did not change the apparent abundance or the luminal accumulation of Na,K-ATPase complexes in the brush border membrane. Furthermore, preliminary in vivo deglycosylation of the luminal membrane by intraventricular injection of a deglycosylation mix consisting of PNGase F, O-glycosidase, neuraminidase, β1–4 galactosidase, and β-N acetylglucosaminidase failed to change membrane targeting of the Na,K-ATPase complex (Helle Damkiers personal observations). This suggests that luminal accumulation of Na,K-ATPase complexes in the CPE does not rely on the heavily glycosylated Na,K-ATPase β2 as suggested in other cell systems [10,11]. Thus, the glycan-crosslinking hypothesis for luminal membrane accumulation seems insufficient in the CPE. 

Ankyrin-3 was mainly accumulated in relation to the luminal membrane of the CPECs, as demonstrated previously [14,15,16]. Clear immunoreactivity was observed along the length of the brush border in a pattern similar to the central core of actin. This is not surprising as long as Na,K-ATPase α1 is known to bind the actin cytoskeleton through ankyrin as described above. Ankyrin-3 and Na,K-ATPase subunits also co-migrated on native PAGE and co-fractionated on sucrose gradients in both plasma membrane fractions and intracellular structures. Interestingly, the Na,K-ATPase α2 and β1 subunits seemed only to come into close proximity with ankyrin-3 at intracellular sites. This may reflect that plasma membrane bound Na,K-ATPase complexes bind to ankyrin-3 via the Na,K-ATPase α1 subunit as in other epithelia. Our RT-PCR analysis confirmed our previous reports of spectrin expression in the CP [15] by the detection of α2-spectrin, along with β1-, β2-, and β3-spectrin. We also verified that α2 and β2 spectrins are partially associated with the luminal membrane domain in this tissue, although most immunoreactivity is observed in the basolateral membrane domain. The apparent lack of spectrins within the brush border is intriguing as these proteins normally serves to attach ankyrin to actin. Both the present and our previous study failed to immunolocalize spectrin forms in the brush border in the CPECs, and both the Na,K-ATPase α1 co-immunoprecipitation and the native PAGE gel cut-outs analyses also suggested a lack of abundant spectrin in relation to the Na,K-ATPase complex in this tissue. This opens for the possibility that the Na,K-ATPase is attached to the cytoskeleton through another actin binding protein such as either indirectly via ankyrin-3 or directly via Na,K-ATPase β1. 

Na,K-ATPase α1 co-immunoprecipitation analysis identified several candidates for the alternative Na,K-ATPase anchoring of which annexin-2, the syntrophins, and plakoglobin/γ-catenin are the most interesting. Among these, annexin-2 belongs to the spectrin gene family and might substitute for α- and β-spectrin in anchoring the Na,K-ATPase complex to the brush border actin. Opposite spectrin that binds Na,K-ATPase α1 via ankyrin-3, annexin-2 directly binds Na,K-ATPase β1 in MDCK cells [22], and potentially serves to stabilize Na,K-ATPase complexes in the brush border in CPECs. As part of an alternative suggestion, the Na,K-ATPase β1 subunits interacts in multiprotein complexes with syntrophin in astrocytes [23], but unfortunately, attempts to immunolocalize annexin-2, the syntrophins, and γ-catenin were unsuccessful. Thus, further studies are warranted in order to directly investigate which of the identified actin binding proteins that tether the Na,K-ATPase complex to the microvillar core of actin. 

Many of the transport systems involved in CP CSF secretion were previously shown to be significantly altered in an Ncbe ko mouse model with respect to both membrane targeting and accumulation [5,24,25]. This was most pronounced for Na,K-ATPase subunits, AQP1, NHE1 and AE2, whereas NKCC1 was unaffected by the loss of Ncbe. Here, we found that genetic deletion of other central transport mechanisms in the CP, such as NKCC1, AQP1, and NHE1, did not affect the abundance of the other transporters in CSF secretion. This is similar to what we previously reported for NBCe2 knockout mice [26]. However, a small but detectable fraction of the Na,K-ATPase subunits and of AQP1 was observed in the basolateral membrane domain of CPECs from NKCC1 ko mice. This indicates a dependence on NKCC1 expression for fully effective luminal membrane targeting and/or accumulation of Na,K-ATPase subunits and AQP1 in the CPECs. Thus, these proteins appear to converge at some point of the synthesis and trafficking pathways. The basolateral targeting of Na,K-ATPase subunits in the NKCC1 ko CP could reflect either (1) a delay in a normally rapid and efficient endocytosis of basolateral Na,K-ATPase, (2) partial insufficiency in a normally strict luminal membrane targeting of the ATPase from the Golgi apparatus. Alternatively, NKCC1 may help sequestering ankyrin-3 to the brush border membrane in the CP necessary to support the luminal anchoring of Na,K-ATPase complexes and AQP1 in at this site.

## 4. Materials and Methods 

### 4.1. Animals

For normal tissue, male C57BL/6 mice were used at age 9–14 weeks (Taconic, Carlsbad, CA, USA). Where indicated, the following genetically modified mouse models were applied: NKCC1 knockout [27], AQP1 knockout [28], NBCn1 knockout [29], and β2/β1 knock-in mice [30]. All procedures conformed to Danish animal welfare regulations, i.e., the Animal Experiments Inspectorate, Ministry of Food, Agriculture, and Fisheries (j.n. 2012-15-2935-00004). 

### 4.2. Fluorescence-Activated Cell Sorting (FACS) of CPECs

Mice were euthanized under isoflurane anesthesia, and the CP from all four ventricles of six mice were dissected and collected in 4 °C HBS (in mM: 145.0 Na^+^, 3.6 K^+^, 1.8 Ca^2+^, 0.8 Mg^2+^, 138.6 Cl^−^, 0.8 SO_4_^2−^, 5.5 Glucose, 10.0 HEPES, 2.0 PO_4_^2−^, with pH 7.4). The pooled CP tissues were incubated in 50 µg/mL Concanavalin A fluorescein (Vector Laboratories, Oxfordshire, GB) in HBS for 10 min at 37 °C, washed and digested in 2 μg/mL dispase (Invitrogen, Carlsbad, CA, USA) and 2 μg/mL collagenase B (Roche, Basel, Switzerland) in calcium-free HBS for 30 min at 37 °C, and incubated in 1:1 mixture of TrypLE Select Enzyme (Thermo-Fisher, Waltham, MA, USA) and cell culture trypsin/EGTA (Thermo-Fisher) supplemented with 1 mg/mL DNase (Sigma, St. Louis, MA, USA) for 10 min at 37 °C. The preparation was inspected on the microscope, passed through a 50 µm filter, and incubated with propidium iodide added before FACS for exclusion of dead cells. Cells were sorted into fluorescein positive and fluorescein negative samples by 4-way purity sorting on a FACSAria III (BD Biosciences, San Jose, CA, USA). The yield and quality control for the FACS isolation of the used CPECs was reported previously [6].

### 4.3. RNA Purification and Reverse Transcriptase Polymerase Chain Reaction (RT-PCR)

After FACS, the collected cells and control tissues were processed for total RNA isolation (Ambion^®^ (Austin, TX, USA) RiboPure™ Kit, Invitrogen) according to the manufacturer’s protocol. Purified mRNA was DNase treated (DNase I, Invitrogen) before cDNA synthesis (SuperScript™ II RT, Invitrogen), and removal of RNA complementary to the cDNA (RNase H, Invitrogen). For the polymerase chain reactions, HotStarTaq Polymerase (Qiagen, Hilden, Germany) was used, and the reactions mixed according to the manufacturer’s instructions (primer pairs are listed in Table 6). Reaction temperatures and periods: 15 min initiation at 95 °C, 30 cycles of amplification [30 s melting at 95 °C, 30 s annealing at 60 °C, and 45 s synthesis at 72 °C], and 10 min termination at 72 °C. Products were mixed with Gel Loading Blue (New England Biolabs, Ipswich, MA, USA) and run on 1–2% Tris/Borate/EDTA gels containing ethidium bromide and imaged on a benchtop UV transilluminator (UVP, AH diagnostics, Tilst, Denmark).

### 4.4. Tissue Fixation and Immunohistochemistry (IHC)

Mice were perfusion fixed via the heart with 4% paraformaldehyde in a phosphate-buffered salt solution (PBS, in mM: Na^+^ 167.2, Cl^−^ 150.0, HPO_4_^2−^ 7.2, H_2_PO_4_^−^ 2.8, pH 7.4) The brains were removed, post-fixed for 2 h, stepwise dehydrated in EtOH and xylene, and embedded in paraffin wax, enabling 2 μm sectioning using a rotary microtome (Leica, Wetzlar, Germany). The sections were de-waxed and stepwise rehydrated, before epitopes were retrieved by boiling the sections in 10 mM Tris buffer (pH 9) with 0.5 mM EGTA. The epitopes were quenched with 50 mM NH_4_Cl in PBS, and unspecific binding was blocked by washing with 1% BSA in PBS with 0.2% gelatin and 0.05% saponin. Sections were incubated overnight at 4 °C with primary antibody diluted in 0.1% BSA and 0.3% Triton X-100 in PBS. Primary antibodies are listed in Table 7. Positive control tissues included kidneys and intestines (not shown).

For visualization, AlexaFluor 488- or 555-coupled donkey anti-rat, -rabbit, or -mouse secondary antibodies (Invitrogen) were used, and cell nuclei were visualized using Topro-3 counterstaining (Invitrogen). Sections were mounted with a coverslip in Glycergel antifade medium (Dako, Glostrup, Denmark) and analyzed using a Leica DMIRE2 inverted microscope with a TC5 SPZ confocal unit using ×63/1.32 NA or ×100/1.4 NA HCX PI Apo oil objectives with 12-bit depth. Sections from at least three mice were analyzed for each labelling. Differential interference contrast (DIC) images were also obtained. Raw images were background subtracted and combined by the use of Image-Pro software (Media Cybernetics). 

### 4.5. Proximity Ligation Assay

Immunolabelling was performed as described above, but with the exception that PLA- and PLA+ probes were applied instead of conventional secondary antibody. The rolling DNA synthesis and labelling with orange fluorescence detection reagent were performed according to the manufacturer’s protocol (Duolink, Sigma). Images were acquired as described above using 543 nm laser excitation and recording the light emission in the range 567–650 nm (averaging 4 images). The fluorescence signal was overlaid onto the corresponding DIC image. Representative micrographs from 3–5 animals are shown. All images with clear separation of reaction products and a suitably clear corresponding DIC image were used for semi-automated quantitation. Reaction products were counted in each of the sites indicated on the bar graphs: Ventricle lumen, luminal membrane domain, cytosol, nucleus, basal membrane domain, and underlying connective tissue.

### 4.6. Stimulated Emission Depletion (STED) Microscopy

Tissue sections were placed on Poly-L-Lysine coated coverslips (Marienfeld, 170 μm) and only the detection steps were modified from the IHC protocol above. For visualization, goat anti-mouse STAR 440SX and goat anti-rabbit STAR 488 (Abberior, Germany) secondary antibodies were used. Sections were mounted with a glass-slide in Prolong Gold Antifade Mountant (Thermo Fisher Scientific). STED images were recorded on a Leica TSC SP8 STED microscope (Manheim, Germany), with 500 nm excitation from a white light laser (for Abberior STAR 488) and at 458 nm from an Argon laser (for Abberior STAR 440SXP). A 592 nm continuous wave laser was used for depletion for both channels. Emission was recorded at 510–560 nm by a gated hybrid detector (0.3 ns) in counting mode (Abberior STAR 488) and at 500–550 nm by a non-gated hybrid detector in counting mode (Abberior STAR 440SXP). STED images were cross-talk corrected and deconvolved using Huygens™ software (Hilversum, The Netherlands). The channels were thresholded and color channels merged in Image J software. Colocalization analyses of STED images was performed by use of Imaris software (Bitplane, Zürich, Switzerland).

### 4.7. Image Analysis

Protein abundance was investigated by quantifying the immunofluorescence intensities from confocal micrographs. All tissues were carefully handled in parallel from the time of fixation throughout embedding, sectioning, staining, and imaging. To avoid saturation of the photomultiplier, the intensity dynamic range (gain and offset) was adjusted to span the intensities of the most intense sample for each antibody. Images were acquired in the focal plane with the highest signal intensity using fixed settings for magnification, laser power, gain, image depth, offset, and averaging for all images with a given antibody.

The immunofluorescence intensities of the stained tissue were quantified from gray-scale images using Image-Pro (Media Cybernetics). For each image, the area of interest was manually defined to avoid counts from non-choroidal tissue or artifacts. A binary mask of the total area of interest was produced from the fluorescence image. Finally, the minimal value for each image pixel was obtained allowing calculation of the total fluorescence count above background within the area of interest (the specific epithelial immunolabeling). For all quantifications, the fluorescence signal was normalized to cell numbers by counting nuclei within the area of interest. All analyzed images were from fourth ventricle CP. In scatter plots, data are normalized to the mean wild type fluorescence signal. 

Assessments of the cross-cellular immunolabeling profile was performed by line profiles (Plot profile feature of the ImageJ software, National Institutes of Health). The line was perpendicular to the basement membrane and expanded to a band with a width of 11 pixels and placed in the mid--nuclear plane of each cell (2 images per mouse). The number of pixels across the cells were compressed to 50 for observation and the average staining intensity for each bin was calculated for each of the mice, where bin #1 represents the basal end of the cells and bin #50 marks the luminal pole. 

### 4.8. Co-Immunoprecipitation

Coupling of the bait antibodies to epoxy beads and the following immunoprecipitation (IP) were performed according to the manufacturer’s protocol (Dynabeads^®^ Co-Immunoprecipitation Kit, Thermo Fischer Scientific). CP from all four ventricles of 10 mice were collected in HBS and lysed in a 1:9 *w*/*v* ratio of tissue to CHAPS buffer (4% CHAPS, 0.1% BSA, 150mM NaCl, 25mM HEPES, pH 7.5) or in the manufacturer’s Extraction Buffer both added 100 mM NaCl, 2 mM CaCl and 2 mM MgCl, 4 µg/mL Leupeptin, 100 µg/mL PhefaBlock, and PhosSTOP (Roche). The tissue was homogenized and incubated end-over-end for 20 min at 4 °C. After centrifugation at 2600× *g* for 5 min at 4 °C, the supernatant was added to the antibody-coupled beads, and incubated by end-by-end rotation for 60 min at 4 °C. The beads were gently washed three times in CHAPS or Extraction buffer and once in PBS. Bound proteins were eluted with Elution Buffer with the above-mentioned protease and phosphatase inhibitors according to the manufacturer’s instructions.

### 4.9. Separation of Subcellular Elements by Differential Centrifugation 

Isolated mouse CP were homogenized in extraction buffer (120 mM NaCl, 25 mM KC1, 10 mM Tris-HC1, 2 mM EDTA, 2 mM EGTA, 0.1 mM DTT, 0.5 mM PMSF, 0.5% Triton X-100, pH 7.5) and loaded onto 5–20% and 5–30% sucrose gradients (in 150 mM NaCl, 15 mM Tris-HCl, 2 mM EDTA, 0.1 mM DTT, pH 8). Samples were centrifuged at 48,000× *g* for 15 min to separate the soluble and insoluble fractions. Linear sucrose gradients were overlaid with 200 µL soluble fraction and centrifuged at 230,000× *g* for 4 h, at 4 °C. Gradients samples were harvested as 25 fractions from top to bottom.

### 4.10. Sodium Dodecyl Sulphate (SDS) and Native Polyacrylamide Gel Electrophoresis (PAGE)

Dissected mouse CP were dissolved in ice-cold dissection buffer containing 0.3 M sucrose, 25 mM imidazole, 1 mM EDTA, 8.4 μM leupeptin (Calbiochem), and 0.4 mM Pefabloc (Roche), with pH 7.2, and sonicated using a probe sonicator (BioLogics Inc. 150 V/T, 3 × 5 bursts at 60% power). For SDS-PAGE, protein contents were quantified (Pierce BCA Protein Assay Kit) and samples were adjusted to 1.5% (*w*/*v*) sodium dodecyl sulfate, 40.0 mM 1,4-dithiothreitol, 6% (*v*/*v*) glycerol, and 10 mM Tris, pH 6.8 with bromophenol blue. The samples were heated at 65 °C for 15 min and approximately 10 µg of protein per sample was separated by 12.5% polyacrylamide gel electrophoresis.

For native PAGE, homogenates were added protease inhibitors (Pefablock, Leupeptin, Sigma-Aldrich) and non-denaturing gel sample buffer, 1% digitonin, and sample additive (NativePage Sample Prep Kit, Invitrogen). After centrifugation at 20,000× *g* for 30 min at 4 °C, soluble protein samples were loaded onto 3–12% Bi-Tris gels (NativePAGE, Invitrogen) and electrophoresed at 150 V for 1 h at 4 °C according to the manufacturers protocol. 

### 4.11. Immunoblotting 

PAGE gels were electrotransferred onto nitrocellulose or PVDF membranes (Ambion). The membranes were blocked with 5% milk in PBS-T (PBS with 0.1% *v*/*v* Tween) and incubated overnight at 4 °C with primary antibody (Table 2) in PBS containing 1% bovine serum albumin (BSA) and 2 mM NaN_3_. After extensive washing in PBS-T, the blots were incubated with horseradish peroxidase-conjugated anti-rabbit secondary antibody (Dako), washed again in PBS-T, and developed with ECL before imaging (ImageQuant LAS4000, GE Healthcare, Chicago, IL, USA). The CP from one mouse only yields in the range 60–100 µg of protein sample. To increase the number of blots per experimental animal, the membranes were divided for high-, medium-, and low-molecular proteins prior to antibody incubation. For dual immunoblotting native PAGE, secondary antibodies were coupled to either 700 nm or 800 nm IR Dye and imaged on a LiCor Odyssey Imager (LiCor Biosciences, Lincoln, NE, USA).

### 4.12. Mass Spectrometry Analysis

IP protein samples were separated by SDS-PAGE, and 8–10 fractions were cut from each gel lane. Each fraction of the samples was reduced, alkylated, digested (trypsin, Promega), and desalted on C18 columns, as described before [33]. MS analysis was performed by a nano Liquid-Chromatography (nLC) (EASY-nLC 1000, Thermo Fisher Scientific, Waltham, MA, USA) coupled to a mass spectrometer (Q Exactive, Thermo Fisher Scientific) through an EASY-Spray nano-electrospray ion source (Thermo Fisher Scientific). A pre-column (Acclaim^®^ PepMap 100, 75 μm × 2 cm, nanoviper fitting, C18, 3 μm, 100 Å, ThermoFisher Scientific) and analytical column (EASY-Spray Column, PepMap, 75 μm × 15 cm, nanoviper fitting, C18, 3 μm, 100 Å, Thermo Scientific) were used to trap and separate peptides, respectively. For nLC separation, buffer A was 0.1% FA and buffer B was 100% ACN/0.1% FA. A 30 min gradient of 5% to 30% buffer B was used for peptide separation. Mass spectrometry constituted of full scans (m/z 300–1800) at a resolution of 70,000 followed by top-10 MS/MS scans at a resolution of 17,500. HCD collision energy was 23–28%. Dynamic exclusion of 30 s as well as rejection of precursor ions with charge state +1 and above +8 were employed.

### 4.13. MS Data Analysis

MS raw files were searched against a mouse protein database (RefSeq database downloaded 13 October 2014, containing 58513 sequences) using both SEQUEST (embedded in Proteome Discoverer 1.4, Thermo Scientific) and MASCOT (version 2.5, linked to Proteome Discoverer 1.4 as well). Precursor and fragment mass tolerances were 10 ppm and 0.02 Da respectively; tryptic peptides with at most two missed cleavage sites were considered. N-terminal acetylation and methionine oxidation were set as dynamic modifications, while cysteine carbamidomethylation was set as static modification. False discovery rate (FDR) was calculated using Percolator, and only rank 1 and high confidence peptides (with a target FDR q-value below 0.01) were included in the final results. 

### 4.14. Statistics

Two-tailed t-tests were used to compare changes between groups (InStat, GraphPad Software, San Diego CA, USA). Levels of *p* < 0.05 were considered adequate to indicate statistical significance.

## 5. Conclusions

We have provided evidence that the Na,K-ATPase complex in the luminal membrane of the mouse CP consists of multiple subunits (at least containing Na,K-ATPase α1, α2, β1, β2 and FXYD1) that probably interact directly. The brush border stabilization of these Na,K-ATPases does not require the expression of the more heavily glycosylated β2 subunit. The Na,K-ATPase complexes may stabilize in the brush border membrane via anchoring through ankyrin-3 or directly via one of the newly identified candidate actin binding proteins. A minor fraction of Na,K-ATPase complexes co-exist with AQP1, NKCC1, AE2, and Ncbe in intracellular structures that are detected by co-immunoprecipitation and PLA. Although NKCC1 did not co-migrate with the Na,K-ATPase complex in native PAGE, the protein seems necessary for full luminal membrane accumulation of the Na,K-ATPase. This might result from a specific insufficiency in the sorting machinery removing the ATPases from the basolateral membrane or by a weakened luminal membrane stabilization for the Na,K-ATPase complex in the absence of NKCC1. One might speculate that Na,K-ATPase complexes in the normal CP also target the basolateral membrane en route to the final position in the luminal membrane.

## Figures and Tables

**Figure 1 ijms-22-01569-f001:**
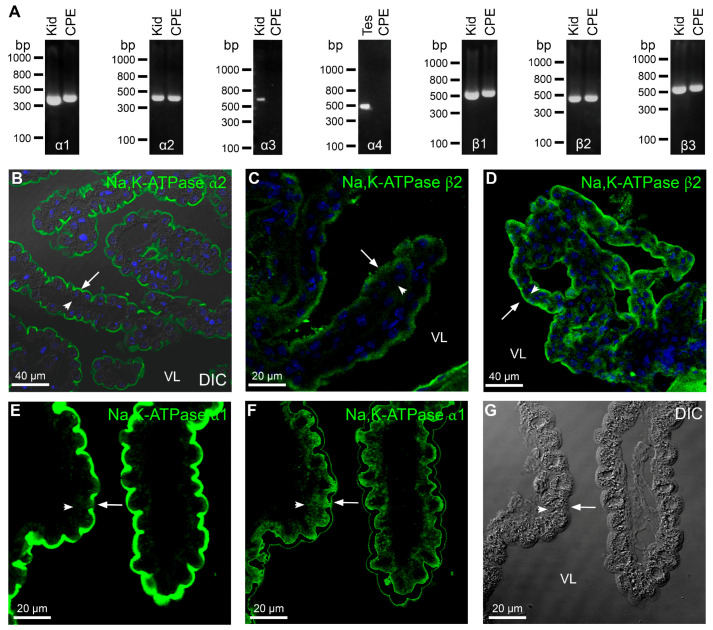
Na,K-ATPase expression in the mouse choroid plexus epithelium (CPE). (**A**) RT-PCR analysis of Na,K-ATPase subunit in fluorescence-activated cell sorting (FACS) isolated CPE cells (CPECs). Intact mouse CP were labelled with concanavalin A-fluorescein from the luminal side. The cells were separated and CPECs enriched by FACS to more than 99.9% purity. After reverse transcription, primers were applied with the cDNA in PCR reactions as indicated on the figure: Na,K-ATPase subunits (α1–4, β1–3). CPE designates FACS isolated CPECs, while Kid and Tes denote the control tissues mouse kidney cortex and mouse testis, respectively. Molecular size is indicated left to the individual gels. (**B**) Immunofluorescence localization of the Na,K-ATPase α2 subunit to the luminal membrane domain of the CP. (**C**,**D**) Micrographs of similar labeling patterns with two antibodies against the Na,K-ATPase β2 subunit. (**E**) Overexposure micrograph of the luminal membrane domain of the Na,K-ATPase α1 subunit (green). (**F**) The same micrograph after thresholding the most intense pixel values to reveal the subcellular sites of low abundance expression of Na,K-ATPase α1. (**G**) Corresponding differential interference contrast (DIC) image of the same cells. Arrows indicate luminal surfaces, while arrowheads mark basolateral surfaces. VL denotes the fourth ventricle lumen.

**Figure 2 ijms-22-01569-f002:**
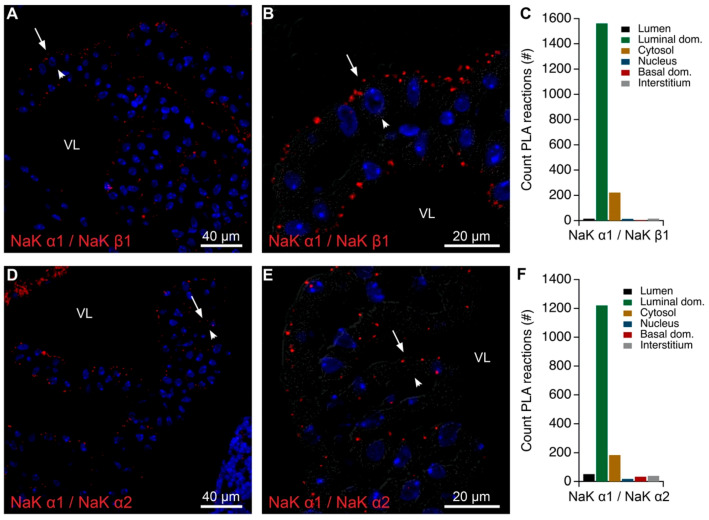
Colocalization of Na,K-ATPase subunits by proximity ligation assay (PLA). (**A**) Representative immunofluorescence image of PLA products (red) using anti-Na,K-ATPase α1 (NaK α1) and anti-Na,K-ATPase β1 (NaK β1) antibodies and cell nucleus counterstain (blue) at low magnification. (**B**) Higher magnification micrograph of the same structure overlaid on the corresponding DIC image. (**C**) Bar graph representing the count of PLA reactions in the indicated image areas. (**D**) Representative immunofluorescence image of PLA products (red) using anti-Na,K-ATPase α1 (NaK α1) and anti-Na,K-ATPase α2 (NaK α2) antibodies and cell nucleus counterstain (blue) at low magnification. (**E**) Higher magnification micrograph of the same structure overlaid on the corresponding DIC image. (**F**) Bar graph representing the count of PLA reactions in the indicated image areas. Arrows indicate luminal surfaces, while arrowheads mark basolateral surfaces. VL denotes the fourth ventricle lumen.

**Figure 3 ijms-22-01569-f003:**
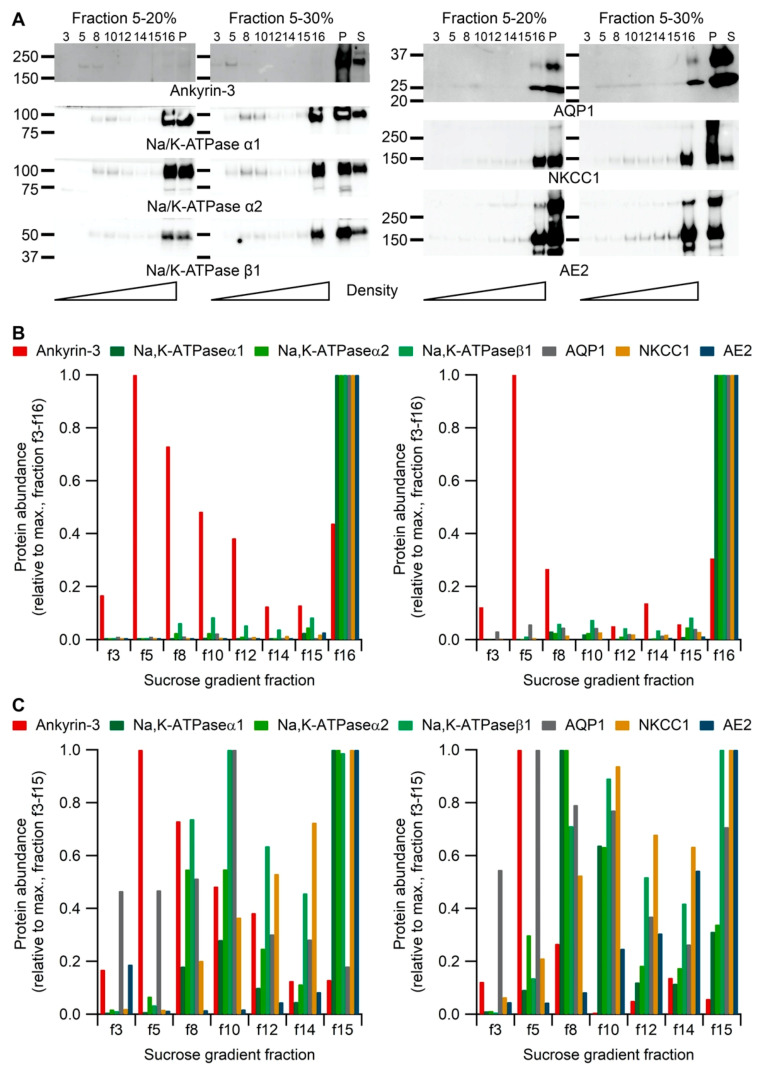
Variation in sub-cellular profiles of CP proteins assessed by differential centrifugation. (**A**) Immunoblots of the sedimentation profiles of CP proteins from 5–20% and 5–30% sucrose gradients. Targeted proteins are indicated below the blots. Molecular sizes are shown on the left (kDa) and fraction numbers are shown above the immunoblots. P: Pellet; S: Supernatant. Below the blots, cartoons are indicating the density from low (left) to high (right). The experiment was repeated for both gradients. (**B**) Densitometric analysis of immunoblots showing each protein abundance relative to the respective maximal abundance observed in fraction 3–16 from 5–20% sucrose gradients (left panel) or from 5–30% sucrose gradients (right panel, *n* = 2). Fraction 16 is plasma membrane enriched. (**C**) Similar analysis showing each protein abundance relative to the respective maximal abundance observed in fraction 3–15 from 5–20% sucrose gradients (left panel) or from 5–30% sucrose gradients (right panel, *n* = 2).

**Figure 4 ijms-22-01569-f004:**
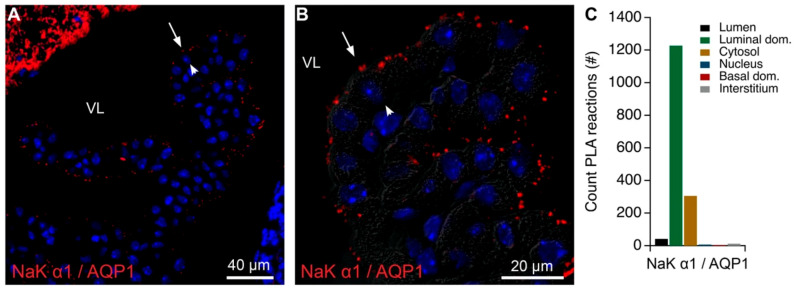
Colocalization of AQP1 and Na,K-ATPase α1 in mouse CPE by PLA. (**A**) Representative immunofluorescence image of PLA products (red) using anti-Na,K-ATPase α1 (NaK α1) and AQP1 antibodies and cell nucleus counterstain (blue) at low magnification. (**B**) Higher magnification micrograph of the same structure overlaid on the corresponding DIC image. Arrows indicate luminal surfaces, while arrowheads mark basolateral surfaces. VL denotes the fourth ventricle lumen. (**C**) Bar graph representing the count of PLA reactions in the indicated image areas.

**Figure 5 ijms-22-01569-f005:**
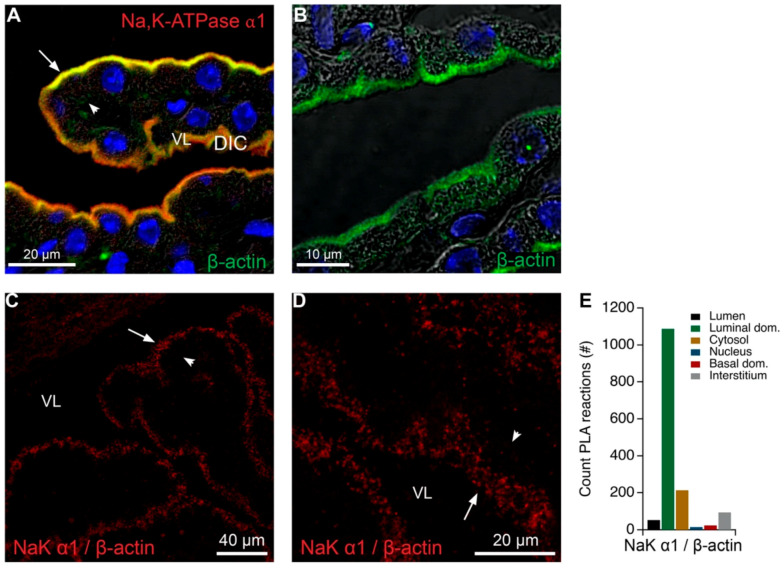
Colocalization of Na,K-ATPase α1 and β-actin in the CP. (**A**) Double immunofluorescence labeling for Na,K-ATPase α1 (red) and β-actin (green) in the CPE. (**B**) Similar labeling for β-actin overlaid on the corresponding DIC image. (**C**) Representative immunofluorescence image of PLA products (red) using anti-β-actin (β-actin) and anti-Na,K-ATPase α1 (NaK α1) antibodies and cell nucleus counterstain (blue) overlaid on the corresponding DIC image. (**D**) Similar micrograph of the same structure at higher magnification. Arrows indicate luminal surfaces, while arrowheads mark basolateral surfaces. VL denotes the fourth ventricle lumen. (**E**) Bar graph representing the count of PLA reactions in the indicated image areas.

**Figure 6 ijms-22-01569-f006:**
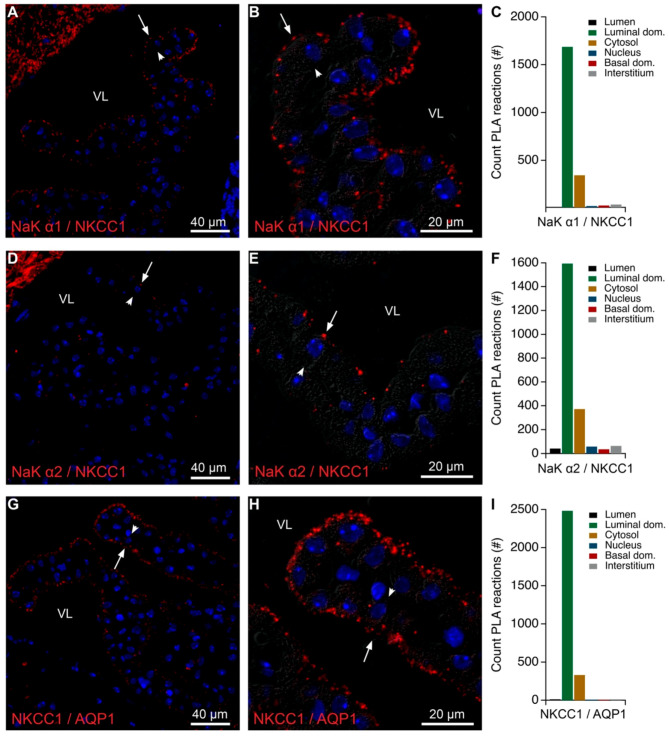
Colocalization of NKCC1 and Na,K-ATPase and AQP1 subunits by PLA. (**A**) Representative immunofluorescence image of PLA products (red) using anti-Na,K-ATPase α1 (NaK α1) and anti-NKCC1 antibodies and cell nucleus counterstain (blue) at low magnification. (**B**) Higher magnification micrograph of the same structure overlaid on the corresponding DIC image. (**C**) Bar graph representing the count of PLA reactions in the indicated image areas. (**D**) Representative immunofluorescence image of PLA products (red) using anti-Na,K-ATPase α2 (NaK α2) and anti-NKCC1 antibodies and cell nucleus counterstain (blue) at low magnification. (**E**) Higher magnification micrograph of the same structure overlaid on the corresponding DIC image. (**F**) Bar graph representing the count of PLA reactions in the indicated image areas. (**G**) Representative immunofluorescence image of PLA products (red) using anti-NKCC1 and anti-AQP1 antibodies and cell nucleus counterstain (blue) at low magnification. (**H**) Higher magnification micrograph of the same structure overlaid on the corresponding DIC image. (**I**) Bar graph representing the count of PLA reactions in the indicated image areas. Arrows indicate luminal surfaces, while arrowheads mark basolateral surfaces. VL denotes the fourth ventricle lumen.

**Figure 7 ijms-22-01569-f007:**
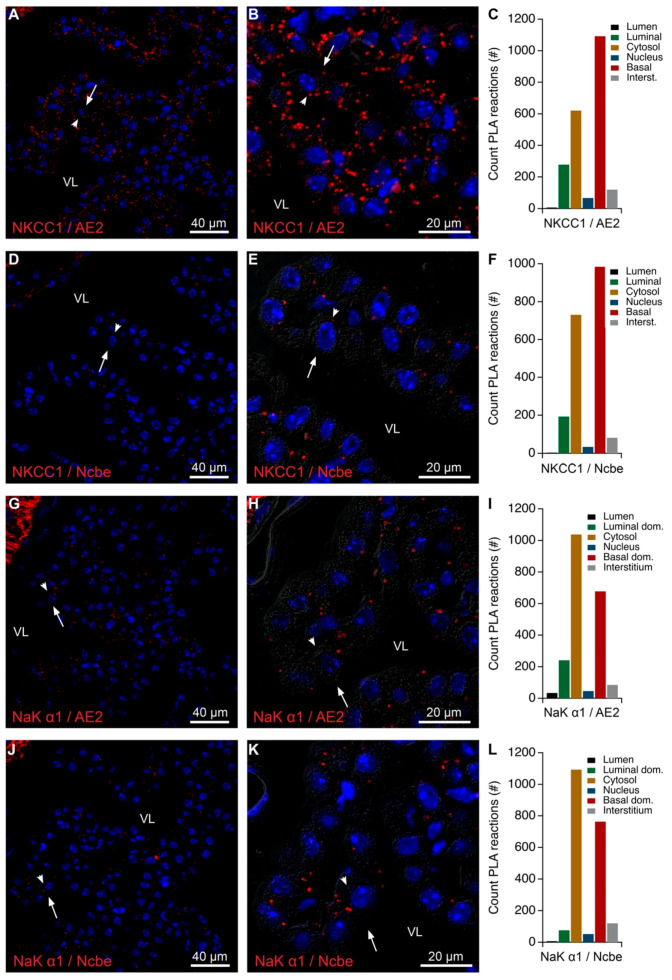
Colocalization of AE2 or Ncbe with NKCC1 or the Na,K-ATPase α1 by PLA. (**A**) Representative immunofluorescence image of PLA products (red) using anti-NKCC1 and anti-AE2 antibodies and cell nucleus counterstain (blue) at low magnification. (**B**) Higher magnification micrograph of the same structure overlaid on the corresponding DIC image. (**C**) Bar graph representing the count of PLA reactions in the indicated image areas. (**D**) Representative immunofluorescence image of PLA products (red) using anti-NKCC1 and anti-Ncbe antibodies and cell nucleus counterstain (blue) at low magnification. (**E**) Higher magnification micrograph of the same structure overlaid on the corresponding DIC image. (**F**) Bar graph representing the count of PLA reactions in the indicated image areas. (**G**) Representative immunofluorescence image of PLA products (red) using anti-Na,K-ATPase α1 (NaK α1) and anti-AE2 antibodies and cell nucleus counterstain (blue) at low magnification. (**H**) Higher magnification micrograph of the same structure overlaid on the corresponding DIC image. (**I**) Bar graph representing the count of PLA reactions in the indicated image areas. (**J**) Representative immunofluorescence image of PLA products (red) using anti-Na,K-ATPase α1 (NaK α1) and anti-Ncbe antibodies and cell nucleus counterstain (blue) at low magnification. (**K**) Higher magnification micrograph of the same structure overlaid on the corresponding DIC image. (**L**) Bar graph representing the count of PLA reactions in the indicated image areas. Arrows indicate luminal surfaces, while arrowheads mark basolateral surfaces. VL denotes the fourth ventricle lumen.

**Figure 8 ijms-22-01569-f008:**
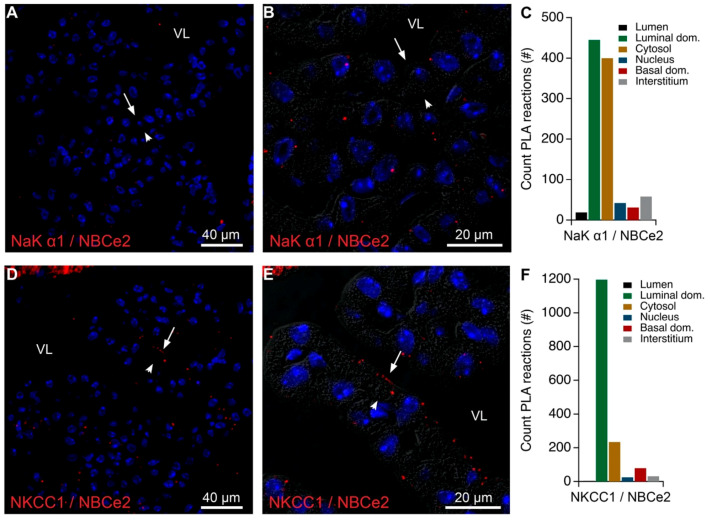
Colocalization of Na,K-ATPase α1 or NKCC1 with NBCe2 by PLA. (**A**) Representative immunofluorescence image of PLA products (red) using anti-Na,K-ATPase α1 (NaKα1) and anti-NBCe2 antibodies and cell nucleus counterstain (blue) at low magnification. (**B**) Higher magnification micrograph of the same structure overlaid on the corresponding DIC image. (**C**) Bar graph representing the count of PLA reactions in the indicated image areas. (**D**) Representative immunofluorescence image of PLA products (red) using anti-NKCC1 and anti-NBCe2 antibodies and cell nucleus counterstain (blue) at low magnification. (**E**) Higher magnification micrograph of the same structure overlaid on the corresponding DIC image. (**F**) Bar graph representing the count of PLA reactions in the indicated image areas. Arrows indicate luminal surfaces, while arrowheads mark basolateral surfaces. VL denotes the fourth ventricle lumen.

**Figure 9 ijms-22-01569-f009:**
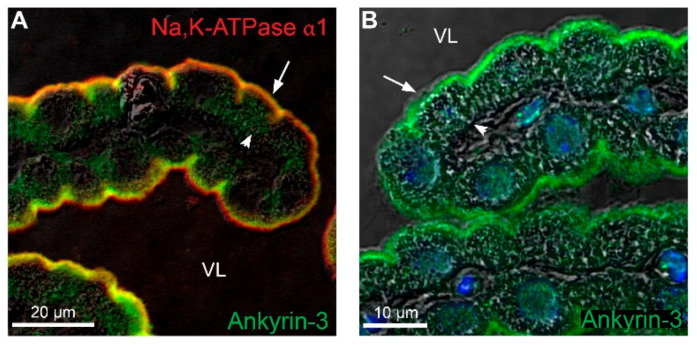
Immunolocalization of ankyrin-3 in the CPE. (**A**) Double immunofluorescence labeling for Na,K-ATPase α1 (red) and ankyrin-3 (green) in the CPE. (**B**) Immunofluorescence labeling for ankyrin-3 (green) overlaid on the corresponding DIC image. Arrows indicate luminal surfaces, while arrowheads mark basolateral surfaces. VL denotes the fourth ventricle lumen.

**Figure 10 ijms-22-01569-f010:**
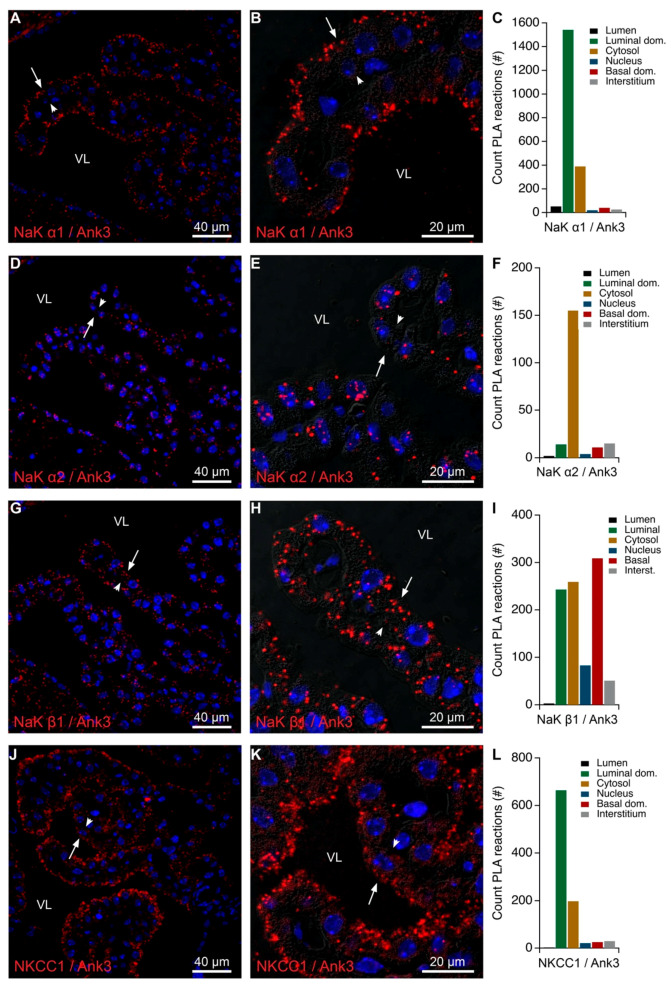
Colocalization of ankyrin-3 and Na,K-ATPase subunits or NKCC1 by PLA. (**A**) Representative immunofluorescence image of PLA products (red) using anti-ankyrin-3 (Ank3) and anti-Na,K-ATPase α1 (NaK α1) antibodies and cell nucleus counterstain (blue) at low magnification. (**B**) Higher magnification micrograph of the same structure overlaid on the corresponding DIC image. (**C**) Bar graph representing the count of PLA reactions in the indicated image areas. (**D**) Representative immunofluorescence image of similar PLA products (red) using a separate set of anti-ankyrin-3 (Ank3) and anti-Na,K-ATPase α2 (NaK α2) antibodies and cell nucleus counterstain (blue) at low magnification. (**E**) Higher magnification micrograph of the same structure overlaid on the corresponding DIC image. (**F**) Bar graph representing the count of PLA reactions in the indicated image areas. (**G**) Representative immunofluorescence image of similar PLA products (red) using anti-ankyrin-3 (Ank3) and anti-Na,K-ATPase β1 (NaK β1) antibodies and cell nucleus counterstain (blue) at low magnification. (**H**) Higher magnification micrograph of the same structure overlaid on the corresponding DIC image. (**I**) Bar graph representing the count of PLA reactions in the indicated image areas. (**J**) Representative immunofluorescence image of PLA products (red) using anti-ankyrin-3 (Ank3) and anti-NKCC1 (NKCC1) antibodies and cell nucleus counterstain (blue) at low magnification. (**K**) Higher magnification micrograph of the same structure overlaid on the corresponding DIC image. (**L**) Bar graph representing the count of PLA reactions in the indicated image areas. Arrows indicate luminal surfaces, while arrowheads mark basolateral surfaces. VL denotes the fourth ventricle lumen.

**Figure 11 ijms-22-01569-f011:**
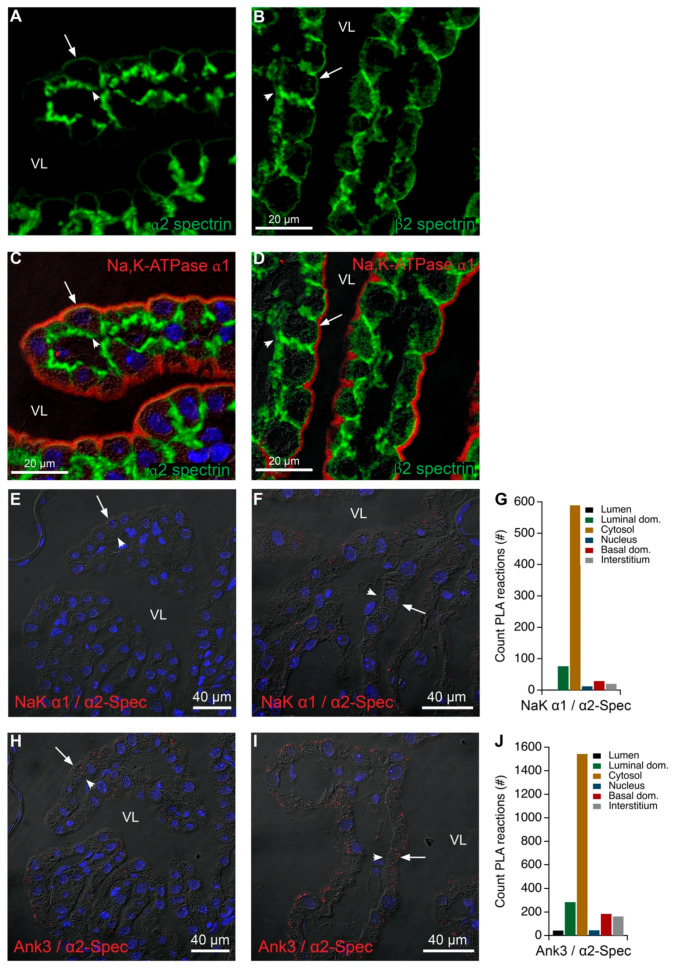
Colocalization of luminal spectrins and Na,K-ATPase α1 or ankyrin-3 in the CP. (**A**) Immunolocalization of α2-spectrin (green) in the CP. (**B**) Similar immunolabeling for β2-spectrin. (**C**) Double immunofluorescence labeling for Na,K-ATPase α1 (red) and α2 spectrin (green) in the CPE (same section as panel/image A). (**D**) Double immunofluorescence labeling for Na,K-ATPase α1 (red) and β2 spectrin (green) in the CPE (same section as panel/image/micrograph B). (**E**) Representative immunofluorescence image of PLA products (red) using anti-α2-spectrin (α2-spec) and anti-Na,K-ATPase α1 (NaK α1) antibodies and cell nucleus counterstain (blue) overlaid on the corresponding DIC image. (**F**) Similar micrograph of the same structure at higher magnification overlaid on the corresponding DIC image. (**G**) Bar graph representing the count of PLA reactions in the indicated image areas. (**H**) Representative immunofluorescence image of PLA products (red) using anti-α2-spectrin (α2-Spec) and anti-ankyrin-3 antibodies (Ank3) and cell nucleus counterstain (blue) overlaid on the corresponding DIC image. (**I**) Similar micrograph of the same structure at higher magnification overlaid on the corresponding DIC image. (**J**) Bar graph representing the count of PLA reactions in the indicated image areas. Arrows indicate luminal surfaces, while arrowheads mark basolateral surfaces. VL denotes the fourth ventricle lumen.

**Figure 12 ijms-22-01569-f012:**
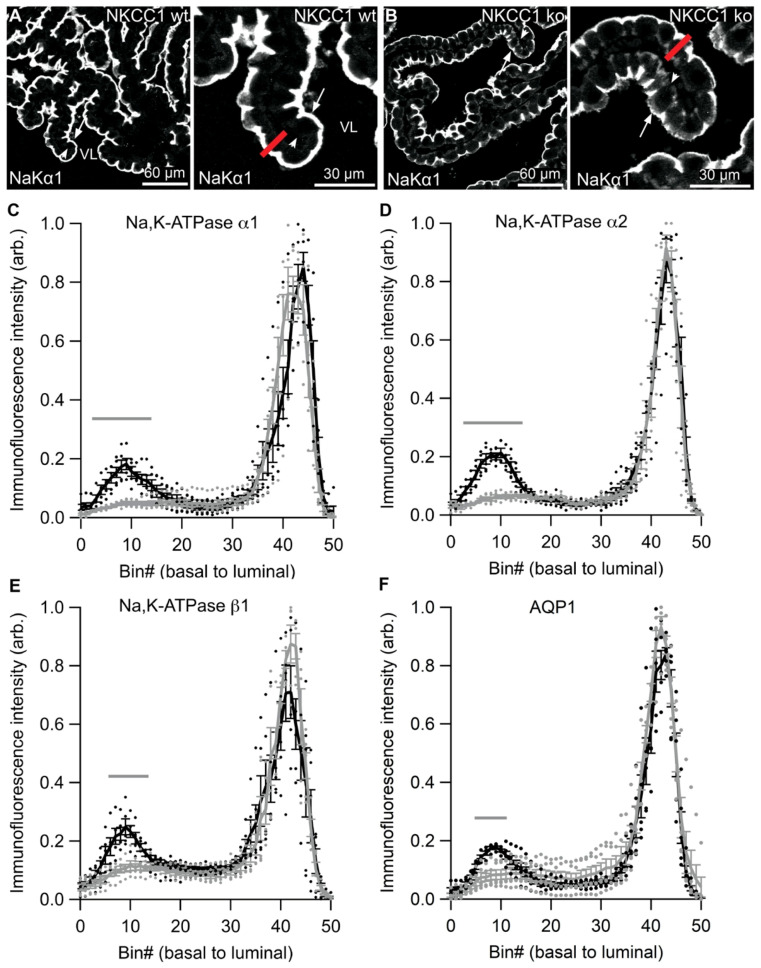
Cross-cellular labeling profiles for Na,K-ATPase and AQP1 in NKCC1 deficient mouse CPECs. (**A**) The Na,K-ATPase α1 (NaK α1) intensity in CPECs from NKCC1 wildtype (wt) mice was quantified by drawing an 11-pixel-wide line (red line) across two cells from each micrograph. The right panel is a higher magnification representation of the left panel. The number of cross cellular basal--to-apical pixels was then compressed to 50 bins for each cell. (**B**) Similar labeling analysis was performed on images of Na,K-ATPase α1 (NaK α1) staining in CPECs from NKCC1 ko mice. Arrows indicate luminal surfaces, while arrowheads mark basolateral surfaces. VL denotes the fourth ventricle lumen. (**C**) Graph showing the average Na,K-ATPase α1 staining intensity (arbitrary units) for each of the 50 bins in CPECs from NKCC1 wt (grey) and ko (black) mice (*n* = 5). (**D**) Similar graphs showing the cross-cellular Na,K-ATPase α2 labeling profiles in CPECs from NKCC1 wt (grey) and ko (black) mice (*n* = 5). (**E**) Graph showing the cross-cellular Na,K-ATPase β1 labeling profiles in CPECs from NKCC1 wt (grey) and ko (black) mice (*n* = 5). (**F**) Graph showing the cross-cellular AQP1 labeling profiles in CPECs from NKCC1 wt (grey) and ko (black) mice (*n* = 5). Bin #1 represents the basal end of the cell, while bin #50 the luminal end. Individual observations are indicated as points, and error bars represent SEM. Grey horizontal bars represent bins with a significant difference in staining intensity between NKCC1 wt and ko mice. *P*-values are given in Appendix A.

**Table 1 ijms-22-01569-t001:** Transport proteins, anchoring proteins, and cytoskeletal proteins co-immunoprecipitated by an anti-,K-ATPase α1 antibody. Maximum sequence coverage (Max. Cover.), Unique peptides (Uniq. Pept.), number of identifications (of five replicates, *n*).

Accession	Description	Abbreviation	Max. Cover.	Uniq. Pept.	*n*
21450277	Sodium/potassium-transp. ATPase a-1	Na,K-ATPase α1	25.71	18	5
6753138	Sodium/potassium-transp. ATPase b-1	Na,K-ATPase β1	20.72	5	5
569003445	Solute carrier family 12 member 2	NKCC1	19.66	17	2
22165384	Tubulin beta-4B chain	β-tubulin	17.08	3	1
6671509	Actin, cytoplasmic 1	β-actin	14.40	4	3
34996479	Phospholemman isoform b	FXYD1	13.04	1	1
6678469	Tubulin alpha-1C chain	α-tubulin	11.80	3	3
30409956	Sodium/potassium-transp. ATPase a-2	Na,K-ATPase α2	11.18	2	1
568960685	Sodium/potassium-transp. ATPase b-3	Na,K-ATPase β3	10.74	2	1
7106439	Tubulin beta-5 chain	β-tubulin	9.91	1	2
226958351	Sodium/potassium-transp. ATPase a-4	Na,K-ATPase α4	8.72	1	1
6680710	Aquaporin-1	AQP1	7.06	2	4
6996913	Annexin A2	Annexin-2	5.31	1	2
7242138	Sodium/potassium-transp. ATPase b-2	Na,K-ATPase β2	4.14	1	1
568933073	Anion exchange protein 2	AE2	3.76	3	1
568971845	Junction plakoglobin	γ-catenin	3.57	2	2
7710096	Beta-1-syntrophin	β1-syntrophin	3.17	1	1
6678059	Beta-2-syntrophin	β2-syntrophin	2.50	1	1
158966733	Inward rectifier potassium channel 13	Kir7.1	2.50	1	1
295054271	Spectrin alpha chain, non-erythroc. 1	α2-spectrin	2.12	4	1
568978640	Alpha-actinin-1	α-actinin	1.79	1	1
334688858	Sodium-driven chloride bicarb. exch.	Ncbe	1.66	1	1

**Table 2 ijms-22-01569-t002:** Coomassie band near the anti-Na,K-ATPase α1 immunoreactive band on native PAGE blot. Maximum sequence coverage (Max. Coverage), Unique peptides (Uniq. Pept.), number of identifications (of five replicates, *n*).

Accession	Description	Abbreviation	Max. Coverage	Uniq. Pept.	*n*
6671509	Actin, cytoplasmic 1	β-actin	33.33	4	1
6678469	Tubulin alpha-1C chain	α-tubulin	29.84	3	2
568907654	Tubulin alpha-4A chain	α-tubulin	26.10	2	1
22165384	Tubulin beta-4B chain	β-tubulin	27.19	11	1
6680710	Aquaporin-1	AQP1	17.84	5	2
158966733	Inward rectifier potassium channel 13	Kir7.1	8.61	3	1
21450277	Sodium/potassium-transp. ATPase a-1	Na,K-ATPase α1	54.06	39	2
30409956	Sodium/potassium-transp. ATPase a-2	Na,K-ATPase α2	48.92	24	2
594190942	Sodium/potassium-transp. ATPase a-3	Na,K-ATPase α3	43.83	19	1
226958351	Sodium/potassium-transp. ATPase a-4	Na,K-ATPase α4	13.57	3	1
6753138	Sodium/potassium-transp. ATPase b-1	Na,K-ATPase β1	38.82	15	2
7242138	Sodium/potassium-transp. ATPase b-2	Na,K-ATPase β2	16.21	5	2
568960685	Sodium/potassium-transp. ATPase b-3	Na,K-ATPase β3	21.90	5	1
37577140	Phospholemman	FXYD1	17.39	2	2

**Table 3 ijms-22-01569-t003:** Co-localization analysis of STED images (NKA: Na,K-ATPase, Spec: spectrin).

Parameter	NKA α1 AQP1	NKA β1 NKA β1	NKA α1 NKA α2	NKA α1 NKA β1	NKA β1 α2-Spec
% colocalized material A thresh.	79.46	41.87	56.1525	31.31	30
% colocalized material B thresh.	39.81	50.34	72.5	45.47	33.93
Pearson’s coefficient in volume	0.659	0.591	0.79953	0.575	0.493
Pearson’s coefficient in ROI	0.659	0.591	0.79953	0.575	0.493
Pearson’s coefficient in volume	0.131	0.224	0.321	0.21	0.457
Thresholded Mander’s coefficient	0.227	0.204	0.24873	0.127	0.09

**Table 4 ijms-22-01569-t004:** Coomassie band at NKCC1, AE2, Ncbe immunoreactive band on native PAGE blot. Maximum sequence coverage (Max. Coverage), Unique peptides (Uniq. Pept.), number of identifications (of five replicates, *n*).

Accession	Description	Abbrev.	Max. Coverage	Uniq. Pept.	*n*
6671509	Actin, cytoplasmic 1	β-actin	4.53	2	1
922959903	Tubulin alpha-4A chain	α-tubulin	6.50	1	1
568933073	Anion exchange protein 2	AE2	8.42	10	4
334688858	Sodium-driven chloride bicarb. exch.	Ncbe	8.37	9	4
124517716	Solute carrier family 12 member 2	NKCC1	25.37	27	4
755513830	Transient receptor pot. cation chan. V4	TRPv4	3.33	3	4
295054266	Spectrin alpha chain, non-erythrocytic 1	α2-spectrin	11.18	16	3
117938334	Spectrin beta chain, non-erythrocytic 1	β2-spectrin	3.76	5	3

**Table 5 ijms-22-01569-t005:** Coomassie band located above the NKCC1, AE2, Ncbe immunoreactive band on native PAGE blot. Maximum sequence coverage (Max. Coverage), Unique peptides (Uniq. Pept.), number of identifications (of five replicates, *n*).

Accession	Description	Abbrev.	Max. Coverage	Uniq. Pept.	*n*
927028891	Actin, cytoplasmic 2	β-actin	8.30	1	1
568933067	Anion exchange protein 2	AE2	15.90	11	4
334688858	Sodium-driven chloride bicarb. exch.	Ncbe	4.32	3	3
755513830	Transient receptor pot. cation chan. V4	TRPv4	4.71	2	2
569003445	Solute carrier family 12 member 2	NKCC1	3.28	2	1
295054271	Spectrin alpha chain, non-erythrocytic 1	α2-spectrin	0.85	1	1
117938334	Spectrin beta chain, non-erythrocytic 1	β2-spectrin	3.85	5	2

**Table 6 ijms-22-01569-t006:** Primers used for RT-PCR with cDNA from FACS purified CPECs (all 5′-3′).

Target cDNA	Forward Primer	Reverse Primer
Na,K-ATPase α1	ATCATCGTAGCCAACGTGCCAG	TGCACTTTAAGAGCGCCGACTC
Na,K-ATPase α2	ACTGCCAGGGGCATTGTGATTG	ACATGTGAGCCACTGTCATGCG
Na,K-ATPase α3	ACGCCCATCGCCATTGAGATTG	TCGCATCAGCTTTACGGAACCC
Na,K-ATPase α4	AGGAGCAAACCACGGGGAAAAC	AAGAAGCAGAACCCCAGCACAC
Na,K-ATPase β1	AAAAGCCAAGGAGGAAGGCAGC	TCGGTTTGAAGCCCAACACTCG
Na,K-ATPase β2	AAAGAGAAGAAGAGCTGCGGGC	TGAACTGGCAGGCACGTTTTGG
Na,K-ATPase β3	TTTTCCCAAACCGCAGACTGCC	ACGTCCCAAGAACTTGTCACGC
Na,K-ATPase β4	GTGGCGGAAATTGCAGATCGTG	ACGGAAGCCTACAATCCGGTTC)
α1-Spectrin	AAGGAAGAGATTGCTGCTCGCC	AAGCCTTCTGCCTCTGAAAGCC
α2-Spectrin	AGCTACCAAACGCAAGCACCAG	TTGACATTCTCCGTTTCGCGGC
β1-Spectrin	AGTGAACCTTGCCGCCAACAAC	TCTCAACATCTGCTTGCCGCTG
β2-Spectrin	AACAACCGCTGGGATGTGGATG	ATCTGGCACCACAGCAGCAATG
β3-Spectrin	CCATTGGTGCGAAGTGCAGAAC	TTGTTTGCGGCACAGCATCC
β4-Spectrin	AACGTGGGGTCACATGACATCG	AAACTTGACGGGCTTCTCCAGC
β5-Spectrin	TGCCTCGCCAAAGATGTGGAAG	TGCTCAATGCGACTGCTGAACC

**Table 7 ijms-22-01569-t007:** Primary antibodies used in this study. IB: Immunoblotting, IP: Immunoprecipitation, IHC: Immunohistochemistry, PLA: Proximity ligase assay, STED: Stimulated emission depletion microscopy.

Target	Antibody	Host	Application	Source
Na,K-ATPase α1	56-0/3B-0	Mouse	IB, IHC, PLA, STED	Gift from Forbush, B., 3rd [31]
Na,K-ATPase α1	05-369	Rabbit	IP	Millipore
Na,K-ATPase α2	16836-1	Rabbit	IB, IHC, PLA, STED	Proteintec
Na,K-ATPase β1	MA3-930	Mouse	IB, IHC, STED	Thermo Scientific
Na,K-ATPase β1	SpET1	Rabbit	PLA, STED	Gift from P Martín-Vasallo [8]
Na,K-ATPase β2	SpET2	Rabbit	IHC	Gift from P Martín-Vasallo [8]
Na,K-ATPase β2	17369 AP	Rabbit	IHC	Genscript
AQP1	2343 AP	Rabbit	IB, IHC, PLA, STED	Genscript
NKCC1 Nt	BSC2-Nt	Rabbit	IB, IHC, PLA, STED	Gift from J Turner [32]
β-actin	B1040	Rabbit	IHC, PLA, STED	Life Sciences
α2-Spectrin	sc-46696	Mouse	IHC, PLA, STED	Santa Cruz
Ankyrin-3	sc-28561	Rabbit	IB, IP, IHC, PLA	Santa Cruz
20S Proteasome	sc-67339	Rabbit	IP	Santa Cruz
AE2	c-terminal	Rabbit	IB, IHC, PLA	Gift from A Stuart-Tilley [14]
NCBE	1139 AP	Rabbit	IB, IHC, PLA	Genscript
NBCe2	9949 AP	Rabbit	IB, IHC, PLA	Genscript

## Data Availability

Not applicable.

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
