# Peer review of "Multiple Na,K-ATPase Subunits Colocalize in the Brush Border of Mouse Choroid Plexus Epithelial Cells"

_ijms, 2021, doi:10.3390/ijms22041569_

Round 1

Reviewer 1 Report

My previous concerns have been addressed by the authors.

Reviewer 2 Report

The authors have addressed most of my concerns. The new quantitative analyses support the original finding and add more strength to the study. I believed that the manuscript has improved scientifically. However, I have the same concern as before, which the fact that the findings of the paper are based on indirect evidence and mostly speculative.

I understand the limitation that the authors are facing (lack proper in vitro systems for the CPE and lack the ability to produce enough proteins in vitro). However, this lessens the impact of the study.

Reviewer 3 Report

Dear authors,

Thank you for taking into consideration all the points I made in your manuscript. I have no further comments in your revised form and I congratulate the authors for the nice article.

This manuscript is a resubmission of an earlier submission. The following is a list of the peer review reports and author responses from that submission.

Round 1

Reviewer 1 Report

Major concern include:

  1. Figure 3 shows the protein levels in different fractions after centrifugation. It is not clear what are those fractions. Could the authors include control marker proteins?
  2. Figure 4, 6, 7 and 10 aim to demonstrate the co-localization of different proteins. Why only one color (red) was used?
  3. To demonstrate that different proteins belong to the same sub-cellular compartment, could the authors use the fractionation approach or another method, rather than just using the images? (Such as results in figure 4, 6, 7 and 10, NKCC1, Ncbe and AE2 co-localizations?) 

Reviewer 2 Report

In this manuscript, Christensen and colleagues are investigating the interaction between Na,K-ATPase subunits and the microvillus cytoskeleton. To so, they perform immunohistochemistry assays, co-IP and native PAGE assays followed by mass spectrometry and proximity ligase assays in mouse choroid plexus epithelium cells (CPECs). This work is rather preliminary and is lacking mechanistic insight.

  1. In this study, the authors have performed FACS to isolate CPECs. The authors have to provide the sorting profile of cell population that has been isolated. In addition, the authors have to evaluate the enrichment in CPECs by quantifying CPECs specific markers.
  2. The data obtained by mass spectrometry have to be confirmed by using antibodies against each protein (AQP1, NKCC1…)
  3. In figure 1, the RT-PCR analysis are missing a housekeeping gene as a control of loading.
  4. The colocalization assays performed by either PLA or double immunofluorescence are lacking quantifications. This would give more strength to the findings of this study.
  5. The authors statement about sucrose gradient assay protein is not accurate (line 169 to 171). The fact that some proteins are abundantly detected in the last fraction can also mean that it did not penetrate the gradient and that they were in fact excluded from it.
  6. In figure 3A, the western blot picture of AQP1 is missing.
  7. The authors claimed that Na,K ATPase subunit are interacting with various protein such AQP1 or ankyrin-3. These interactions need to be confirmed by testing if they interact directly or not (binding assay).
  8. Protein nomenclature is not uniform in the text. Sometimes the authors are using the abbreviation and other times are not.

Reviewer 3 Report

The manuscript “Multiple Na,K-ATPase subunits colocalize in the brush border of mouse choroid plexus epithelial cells” provides evidence of the presence of multiple Na,K-ATPase subunits in the brush border of mouse Choroid plexus. Interesting also suggests that Na,K-ATPase complexes may stabilize in the choroid plexus by anchoring through anchyrin-3.

The study is straightforward and the manuscript comprehensive. The experimental design and techniques are appropriate for the aim of the study. Nevertheless I have minor revisions that the authors used may consider to improve their study.

  1. The study could be highlighted with a paragraph discussing the physiological importance of the presence of multiple Na,K-ATPase subunits in the brush border of the choroid plexus. And also, how the stabilization of Na,K-ATPase subunits is important for the production of CSF.
  2. Introduction, line 62. “(…) Na,K-ATPase in the CP requires a delivery (…)”
  3. Introduction, line 100, line 106 and line 197. Figure S1, Figure S2C and S2D do not match with the description in the text. Please confirm.
  4. Results, line 322, line 366 and line 430. Explanations and interpretations of results should be omitted from the Results section and rather be included in the discussion, together with literature references.
  5. Figure 12 legend. Include in the legend the statistical test used and P value that indicate statistical significance.
  6. Supplemental Figure 1D. If possible, increase image size since bands are difficult to see.